# CONTRASTIVE LEARNING WITH SIMPLICIAL CONVOLUTIONAL NETWORKS FOR SHORT-TEXT CLASSIFICATION

## ABSTRACT

Text classification is a fundamental task in Natural Language Processing (NLP). Short text classification has recently captured much attention due to its increased amount from various sources with limited labels and its inherent challenges for its sparsity in words and semantics. Recent studies have adopted self-supervised contrastive learning across different representations to improve performance. However, most of the current models face several challenges. Firstly, the augmentation step might not be able to generate positive and negative samples that are semantically similar and dissimilar to the anchor respectively. Secondly, the text data could be enhanced with external auxiliary information that might introduce noise to the sparse text data. In addition, they are limited in capturing higher-order information such as group-wise interactions. In this work, we propose a novel document simplicial complex construction based on text data for a higher-order message-passing mechanism. We develop a simplicial complex representation for text sentences based on the directed word co-occurrence. Novel features are proposed for 0-simplex (word), 1-simplex (word-pair), and 2-simplex (three consecutive words) to characterise intrinsic higher-order structural information among words. We also enhance the short text classification performance by contrasting the structural representation with the sequential representation generated by the transformer mechanism for improved outcomes. The proposed framework, Contrastive Learning with Simplicial Convolutional Networks (C-SCN), leverages the expressive power of graph neural networks, models higher-order information beyond pair-wise relations and enriches features through contrastive learning. Experimental results on four benchmark datasets demonstrate the capability of C-SCN to outperform existing models in analysing sequential and complex short-text data.

## 1 INTRODUCTION

Text classification is a fundamental task in Natural Language Processing (NLP). It involves analysing the content of texts and determining which predefined category they belong to based on their representation. Unlike longer texts, short texts have recently captured much attention, with an increase in the number appearing in various sources, such as social media, search snippets, and news feeds. However, these short texts with a few words pose challenges to the current models in generating effective representations and are not usually labelled in real-world cases (Linmei et al. (2019)). Supervised learning on short-text classification has gained significant attention and has been applied to different tasks for web reviews (Pang & Lee (2004)), news feed (Yao et al. (2019)) and medical information (Liu et al. (2020)). On the other hand, data labelling has been expensive, labour-intensive and time-consuming. Few-shot learning has been popular with low resources required by training on a few labelled samples, either with or without pre-training. Furthermore, graph models have been widely used to capture complex relationships between text data's structural, semantic, and syntactic meanings. To address the label scarcity issue, contrastive learning has been adopted to enhance performance. Many researchers (Sun et al. (2022); Wen & Fang (2023); Liu et al. (2024)) have explored the effectiveness of combining graph models and contrastive learning within the scope of few-shot learning.

Although they have achieved successful outcomes, some limitations and challenges still exist. Firstly, the data augmenting step and the negative sampling step of contrastive learning might distort the semantic meaning and introduce unnecessary noise. For example, removing graph components is adopted as a data augmentation strategy, but it might disrupt the text's original meaning. An instance from the Movie Review (MR) dataset (Pang & Lee (2005)): "there's not enough to sustain the comedy" while removing the word "not" reversely changes the meaning of this short sentence. Furthermore, negative sampling of texts with different syntaxes while similar semantics might be designed to be pushed away from each other. Secondly, some auxiliary information such as entities, latent topics, and part-of-speech (POS) tags (such as nouns and verbs) might be added to graph models for language understanding and enriching the limited available local context. However, this step might introduce misinformation, such as pulling documents that express opposite semantics but similar topics closer. Lastly, graph models are mathematically limited in modelling higher-order features, such as group-wise interactions among a few nodes and edges expressed in terms of phrases. For example, the short sentence "It is what it is" uses repetition to emphasise the acceptance of the status quo. At the same time, graph models with only nodes and edges learn pairwise interaction. They need to extend the number of layers in order for words to incorporate the meaning of other words further apart. Group-wise phrase "it is" needs to be linked with "what" to model such repetition.

To address the challenges mentioned above, a novel model combining higher-order features with contrastive learning is proposed in this paper called Contrastive Learning with Simplicial Convolutional Networks (C-SCN) for short-text classification tasks. Specifically, SCN adopts simplicial complex to robustly model richer and more complex information for better document understanding. Document simplicial complexes are firstly constructed based on text data, and respective features are defined for simplexes of different complexities. We further integrate the features into an inductive message-passing mechanism, considering long-range structural information for individual document simplicial complexes. Furthermore, contrastive learning is embraced to compare the structural representation from SCN and sequential representation from the transformer model so that the power of both sides can be combined for better performance in a few-shot learning setting.

Our work's key contributions are as follows. Firstly, we propose the construction of simplicial complexes based on text data and define features on 0-simplexes, 1-simplexes and 2-simplexes in the context of short-text classification in the message-passing mechanism of SCN. Secondly, we extend SCN with contrastive learning, such as C-SCN, where the power of sequential representation from the transformer model is integrated to solve the existing limitations and challenges. Lastly, the experiment with C-SCN in benchmark short text classification tasks demonstrates better results than competitive baseline models in the few-shot setting.

The remainder of the paper is organised as follows: Section 2 reviews the literature on graph neural networks, contrastive learning, and neural networks on simplexes. Section 3 outlines the proposed model structure for message passing on higher-order structures and contextualises the methods in text classification. Section 4 introduces four short text classification task datasets from various domains used in the experiments. Section 5 presents the performance metrics compared with other models and ablation studies. Finally, Section 6 concludes the work and proposes future directions.

## 2 LITERATURE REVIEW

### 2.1 GRAPH NEURAL NETWORKS IN SHORT-TEXT CLASSIFICATION

Graph neural networks (GNN) are powerful deep learning models to model representations of structural data (Scarselli et al. (2009)). Through a message-passing mechanism, features of nodes and edges are aggregated in the neighbourhood formed by components in the local document. Texts could be used to construct different types of graphs, such as heterogeneous graphs (Yao et al. (2019)), knowledge graphs (Ye et al. (2019)), dynamic graphs (Chen et al. (2020)), and hypergraphs (Ding et al. (2020)). Early graph neural networks, such as Graph Convolutional Networks (GCN) (Kipf & Welling (2017)), Graph Isomorphism Networks (GIN) (Xu et al. (2019)) and Graph Attention Networks (GAT) (Veličković et al. (2018)), are integrated with the text graphs for improved results. Recently, model fusion has been adopted by Lin et al. (2021) to jointly train the transformer model BERT with graph models with text data. On the other hand, graph models are limited in modelling higher-order information in the group-wise form.

## 2.2 Contrastive Learning

Unlike traditional supervised learning, where a label is required for training, contrastive learning is a self-supervised technique where augmented views of the same object are used to train a model which could gather positive samples closer and negative samples further apart (Jaiswal et al. (2021)). With downstream tasks, contrastive learning has been actively applied in short-text classification scenarios with few-shot settings. Sun et al. (2022) integrates the heterogeneous graph attention mechanism with neighbouring contrastive learning to enrich the terms beyond the document and extend the relations among documents; Wen & Fang (2023) pre-trains text and graph encoders followed by few-shot and zero-shot fine-tuning process; Liu et al. (2024) innovates in augmented view of graph features through the singular value decomposition (SVD) of the feature matrix and in assigning weak labels to document through $k$-means clustering. On the other hand, these current methods require a large amount of resources in preprocessing in the form of pre-training or enriching text with additional information, such as entity recognition and POS tagging processes. The augmented view in contrastive learning might also introduce unnecessary noise and misleading information.

## 2.3 Topological Deep Learning (TDL)

Topological deep learning combines the techniques from deep learning and topological tools that structure data manifolds (Zia et al. (2024)). Topological representations, including cell complexes (Hajij et al. (2020); Giusti et al. (2023); Bodnar et al. (2022)), simplicial complexes (Bodnar (2022); Schaub et al. (2022)), combinatorial complexes (Hajij et al. (2023)), sheaves (Hansen & Ghrist (2019)) and hypergraphs (Feng et al. (2018); Bai et al. (2021)), model not only pair-wise interactions that are present on a graph, but also higher-order interactions among three elements or more. Algebraic topology-based methods have achieved noteworthy results in protein analysis (Xia & Wei (2014); Sverrisson et al. (2021); Wee & Xia (2022)), virus analysis (Chen et al. (2022)), drug design (Cang & Wei (2017)) and material property classification (Reiser et al. (2022); Townsend et al. (2020)), where topological representations demonstrate their robustness against deformation and noise. Extending from algebraic topology-based methods, TDL employs the message-passing mechanism on higher-order components, where the communication of information has been propagated through any neighbourhood relations (Roddenberry et al. (2021); Bodnar (2022); Hajij et al. (2023)). However, there is a lack of studies on non-time-series sequential analysis with TDL on the text data, and we aim to explore its usage in the new field.

## 3 Methods

### 3.1 Simplicial Convolutional Networks (SCN)

We first provide the necessary details related to constructing document simplicial complexes, followed by the message-passing mechanism on the higher-order structures.

*Definition. (Abstract Simplicial Complex)* An **abstract simplicial complex** is a family of sets $\mathcal{K}$ that satisfies the condition: for any set $\sigma \in \mathcal{K}$, every non-empty finite subset $\sigma' \subseteq \sigma$ must also be in $\mathcal{K}$. Each element of $\mathcal{K}$ is called a **simplex**. A set $\sigma$ is referred to as a $k$-simplex if its cardinality $|\sigma| = k + 1$, denoted as $\sigma_k$. All $(k-1)$-simplexes, $\sigma_{k-1}$, are **faces** of $\sigma_k$ if they are subsets of $\sigma_k$, while all $(k+1)$-simplexes, $\sigma_{k+1}$, are **cofaces** of $\sigma_k$ if they contain $\sigma_k$ as one of their faces.

Denote $\mathcal{K}_k$ the set of $k$-simplexes for $\mathcal{K}$. $\mathcal{K}_0$ will be referred to as the set of **0-simplexes** (nodes). $\mathcal{K}_1$ refers to the set of **1-simplexes** (edges) and $\mathcal{K}_2$ refers to the set of **2-simplexes** ("filled" triangles). For the text classification task, we construct the document as a simplicial complex with initial representations of 0-simplexes, 1-simplexes, and 2-simplexes, as shown in Figure 1. We embrace the bag-of-word model (Harris (1954)) and treat each word and punctuation as distinct 0-simplexes initialised from GloVe embeddings (Pennington et al. (2014)). The three types of direction of 1-simplexes follow the sequential order of the tokens in each text as shown in Figure 1. 2-simplexes are formed when any three words form a "filled" triangle. We differentiate their nine identities by the neighbouring 1-simplexes for the 2-simplexes to be formed. An example of the 1-simplex $e_{st}$ is shown in Figure 2 where the types of 2-simplex formed are determined by the 1-simplex between $s$ and $o$ and the 1-simplex between $t$ and $o$. It is to be noted that the self-loop is not considered part of the 2-simplex formation since we consider unique 0-simplexes appearing in texts. One target

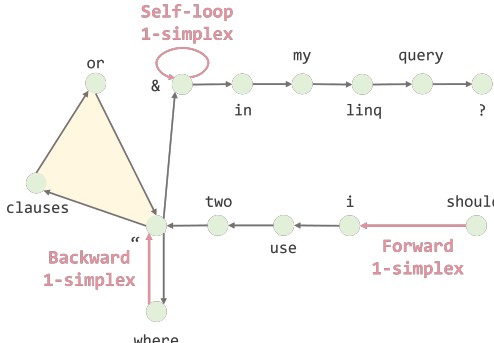

Figure 1: One document simplicial complex constructed for a document example from the Snippets dataset (Phan et al. (2008)) with different types of flow directions. Words and punctuation are tokenised into individual *0-simplexes* (nodes). *1-simplexes* (edges) are formed if 0-simplexes are next to each other with directions in chronological order. Lastly, *2-simplexes* (triangles) are constructed if the three words form a "filled" triangle. Three types of 1-simplexes are illustrated: (1) **Forward 1-simplexes** are the ones following the chronological order which points to the word that first appears in the text; (2) **Backward 1-simplexes** are 1-simplexes pointing to the word which is used before and referenced again; (3) **Self-loop 1-simplexes** are formed when 1-simplexes connect the same word.

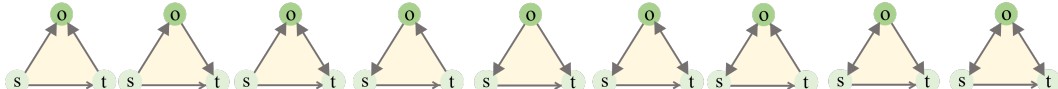

Figure 2: 2-simplexes types for a 1-simplex with source 0-simplex $s$ and target 0-simplex $t$. For the 1-simplex $e_{st}$ with a source 0-simplex $s$, a target 0-simplex $t$ and the defined direction from $s$ to $t$, nine types of 2-simplexes could define the information flow through the 2-simplex. For a 0-simplex $o$ that forms a 2-simplex with the target 1-simplex $e_{st}$, there exist three types of 1-simplexes between 0-simplex $o$ and 0-simplex $s$, as well as between 0-simplex $o$ and 0-simplex $t$: into, out of or bidirectional, resulting in nine types of 2-simplex with respect to the 1-simplex $e_{st}$.

1-simplex could be part of multiple 2-simplexes. The 1-simplex and 2-simplex embeddings will be initialised randomly, and all embedding matrices are optimised during training.

The message-passing mechanism leverages the connectivity information in simplicial complexes. For a simplex $\sigma_k$, we denote its **boundary adjacent simplexes** $\mathcal{B}(\sigma_k)$ as the set of lower-dimensional simplexes $\sigma_{k-1}$ on the boundary of $\sigma_k$, its **co-boundary adjacent simplexes** $\mathcal{C}(\sigma_k)$ as the set of higher-dimensional simplexes $\sigma_{k+1}$ with $\sigma_k$ on their boundaries, its **lower adjacent simplexes** $\mathcal{N}_\downarrow(\sigma_k)$ as those with the same dimension as $\sigma_k$ that share a lower-dimensional simplex $\sigma_{k-1}$ on their boundary, and its **upper adjacent simplexes** $\mathcal{N}_\uparrow(\sigma_k)$ as those with the same dimension as $\sigma_k$ that are on the boundary of the same higher-dimensional simplex $\sigma_{k+1}$ (Bodnar et al. (2021)).

The hidden representation of simplexes will be initialised with vectors $\mathbf{x}_{\sigma_i}$ for all $\sigma_k \in \mathcal{K}_k, k \in \{0, 1, \cdots, K\}$ and we set $K = 2$. That means at the initial state, the layer representation for simplex $\sigma_k$ is $h_{\sigma_k}^{(0)} = \mathbf{x}_{\sigma_k}$ for $k \in \{0, 1, \cdots, K\}$. The messages are then aggregated according to the neighbourhood in which the simplexes sit: at the state $\ell + 1$ and for the target $k$-simplex $\sigma_k$, the message function $M_k^{(\ell+1)}$ collects information from neighbouring simplexes of the same dimension $\sigma_k' \in \mathcal{N}(\sigma_k)$ where $\mathcal{N}(\sigma_k) = \mathcal{N}_\downarrow(\sigma_k) \cup \mathcal{N}_\uparrow(\sigma_k)$, those of one dimension lower $\sigma_{k-1} \in \mathcal{B}(\sigma_k)$ and those of one dimension higher $\sigma_{k+1} \in \mathcal{C}(\sigma_k)$ as illustrated in Equation (1). In the context of text classification, we adopt the message function as a multi-layer perception (MLP) for both 0-simplexes $\sigma_0 \in \mathcal{K}_0$ and 1-simplex $\sigma_1 \in \mathcal{K}_1$ updates. The message passing is set to collect information from neighbouring simplexes and co-boundary adjacent simplexes. For the aggregation of all the components in the document simplicial complex, we adopt row summation as illustrated in Equation (2).

For individual $\sigma_k \in \mathcal{K}_k$ and $k \in \{0, 1, \cdots, K\}$,

$$m_{\sigma_k}^{(\ell+1)} = \underset{\substack{\sigma_k' \in \mathcal{N}(\sigma_k) \\ \sigma_{k-1} \in \mathcal{B}(\sigma_k) \\ \sigma_{k+1} \in \mathcal{C}(\sigma_k)}}{\text{AGG}} \left( \phi \left( M_k^{(\ell+1)}(h_{\sigma_k'}^{(\ell)}, h_{\sigma_{k-1}}^{(\ell)}, h_{\sigma_{k+1}}^{(\ell)}) \right) \right) \tag{1}$$

$$= \sum_{\substack{\sigma_k' \in \mathcal{N}(\sigma_k) \\ \sigma_{k+1} \in \mathcal{C}(\sigma_k) \cap \mathcal{C}(\sigma_k')}} \phi \left( \text{MLP}_k^{(\ell+1)} \left( h_{\sigma_k'}^{(\ell)} + h_{\sigma_{k+1}}^{(\ell)} \right) \right) \tag{2}$$

where for $k = 0$, we do not consider simplex of dimension $(n-1)$, $h_{\sigma_k}^{(\ell)}$ refers to the simplex $\sigma_k$'s feature at the state $\ell$. $h_{\sigma_{k+1}}^{(\ell)}$ is set to a zero vector if $\mathcal{C}(\sigma_k) \cap \mathcal{C}(\sigma_k')$ is empty. $\text{MLP}^{(\ell+1)}$ refers to trainable multi-layer perception at the state $\ell + 1$.

Similarly to the GNN framework, the update function $\text{UPDATE}^{(\ell+1)}$ will synchronise the representation of the $k$-simplex to the new state, as shown in Equation (3), and we adopt the Gated Recurrent Unit (GRU) for the text classification task.

$$h_{\sigma_k}^{(\ell+1)} = \text{UPDATE}_k^{(\ell+1)}(h_{\sigma_k}^{(\ell)}, m_{\sigma_k}^{(\ell+1)}) = \text{GRU}_k^{(\ell+1)}(h_{\sigma_k}^{(\ell)}, m_{\sigma_k}^{(\ell+1)}) \tag{3}$$

Lastly, the readout function READOUT will obtain the representation for the document simplicial complex by pooling $k$-simplexes' features of the final state $L$ in Equation (4). A global self-attention mechanism (Lin et al. (2017)) is specifically applied for text data, summarising the 0-simplexes and 1-simplexes. For the final layer representation $h_{\mathcal{K}}^L$ of the document simplicial complex with 0-simplexes $\sigma_0 \in \mathcal{K}_0$ and 1-simplexes $\sigma_1 \in \mathcal{K}_1$, its individual simplex attention score $\alpha_{\sigma_k}$ is derived with two multi-layer perceptions without bias denoted by $\mathbf{W}_1$ and $\mathbf{W}_2$. The final simplex representation for the document simplicial complex, $h_{\mathcal{K}}^L$, is hence the summation of the attention score multiplied by the respective final simplex features $h_{\sigma_k}^L$ for $k \in \{0, 1\}$.

$$h_{\mathcal{K}}^L = \text{READOUT} \left( \{h_k^{(L)} | k \in \{0, 1, \cdots, K\}\} \right) \tag{4}$$

$$= \left( \sum_{\sigma_0 \in \mathcal{K}_0} \alpha_{\sigma_0} h_{\sigma_0}^L \right) \oplus \left( \sum_{\sigma_1 \in \mathcal{P}_1} \alpha_{\sigma_1} h_{\sigma_1}^L \right) \tag{5}$$

$$\alpha_{\sigma_k} = \frac{\exp \left( \tanh(\mathbf{W}_1 h_{\sigma_k}^L) \cdot \mathbf{W}_2 \right)}{\sum_{\sigma_k' \in \mathcal{K}_0} \exp \left( \tanh(\mathbf{W}_1 h_{\sigma_k'}^L) \cdot \mathbf{W}_2 \right)} \tag{6}$$

where $h_k^{(L)}$ refers to the final collective representation for all $\sigma_k \in \mathcal{K}_k$. Finally, a linear layer with a softmax classifier will transform the results to the same dimension as the label set and make predictions. A summary of the proposed SCN framework is illustrated in Figure 3. With the input sentence "a thriller without a lot of thrills.", a simplicial complex could be constructed with the following components. For 0-simplexes, we have matches $v_1$: "a", $v_2$: "thriller", $v_3$: "without", $v_4$: "lot", $v_5$: "of", $v_6$: "thrills", and $v_7$: ".". For 1-simplexes, $e_1, e_2, e_4, e_5, e_6, e_7$ are forward edges connecting two words, and $e_3$ is a backward edge between "without" and "a". Lastly, a 2-simplex $\tau$ is a type-4 triangle formed by the 1-simplexes connecting among the words "a", "thriller" and "without".

Assuming message functions are fully connected neural networks, the SCN could be evaluated with three components: feature transformation in neural networks, neighbourhood aggregation and non-linear activation. Assuming that all the layers are of the same size $F$ and the embedding size is fixed with $F$ for 0-simplexes, 1-simplexes and 2-simplexes, the features are initialised from all three kinds of simplexes and the dense matrix multiplication takes $\mathcal{O}(|\mathcal{K}_0|F^2 + |\mathcal{K}_1|F^2 + |\mathcal{K}_2|F^2) = \mathcal{O}(|\mathcal{K}|F^2)$. The aggregation and update step will take $\mathcal{O}(|\mathcal{K}_1|F^2 + |\mathcal{K}_2|F^2)$ for 0-simplex and 1-simplex updates. Non-linear activation is an element-wise function which will take $\mathcal{O}(|\mathcal{K}_0| + |\mathcal{K}_1|)$. As a result, over $L$ layers, the final time complexity is $\mathcal{O}(|\mathcal{K}_0| + |\mathcal{K}_1| + |\mathcal{K}_1|F^2 + |\mathcal{K}_2|F^2 + |\mathcal{K}|F^2) = \mathcal{O}(|\mathcal{K}|F^2)$.

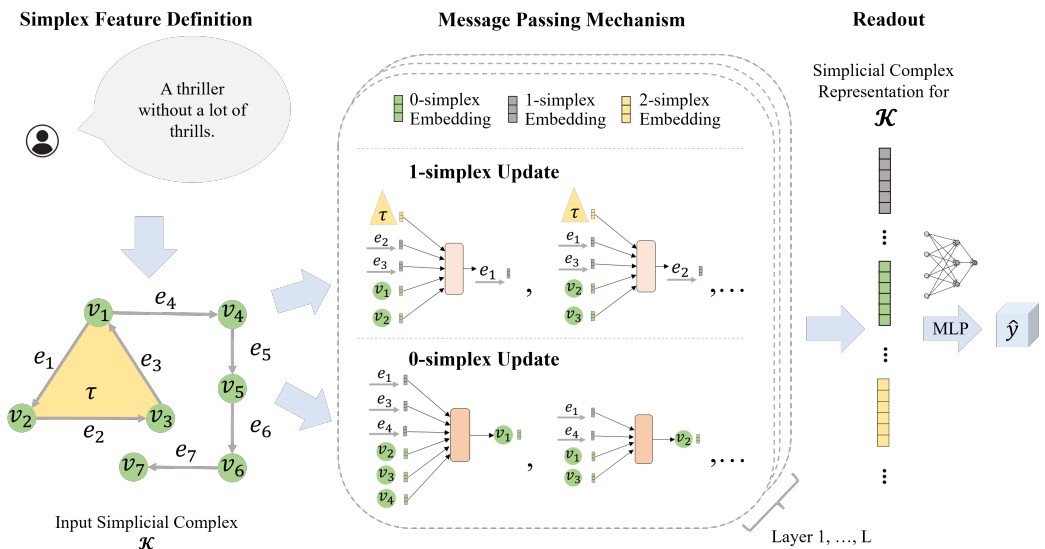

Figure 3: Message-passing mechanism in **Simplicial Convolutional Networks (SCN)** up to two-dimension. The above figure illustrates an example of a simplicial complex for a seven-token sentence with a message-passing mechanism that collects neighbouring information from the same dimension, one dimension lower and one dimension higher. The input simplicial complex $\mathcal{K}$ consists of 0-simplexes $v_1, \cdots, v_7$, 1-simplexes $e_1, \cdots, e_7$ and the 2-simplex $\tau$. Pre-defined and trainable features of different simplexes are used as input on the left-hand side. SCN leverages neighbouring, boundary and co-boundary simplexes to carry feature information and update the target 0-simplexes and 1-simplexes separately in different layer states. Finally, the features of different simplexes are read out for downstream tasks.

### 3.2 SCN WITH CONTRASTIVE LEARNING

To alleviate the abovementioned challenges with contrastive learning, we adopt a dual-encoder framework inspired by Wen & Fang (2023) where we generate text representations from transformer blocks and graph representations from SCN in parallel; hence, the training process could optimise the contrastive learning and classification task as shown in Figure 4.

We employ the BERT model (Devlin et al. (2019)) as the text encoder and SCN encoder that digest the document data $\mathbf{x}_{\text{doc}}$. We denote $Z_t, Z_s \in \mathbb{R}^\gamma$ as the text encoder and SCN encoder output. MLP($\bullet$) is a linear layer that processes the output to the target space's dimension $\gamma$.

$$Z_t = \text{MLP}_t(\text{BERT}(\mathbf{x}_{\text{doc}})), \ Z_s = \text{MLP}_{sc}(\text{SCN}(\mathbf{x}_{\text{doc}})) \tag{7}$$

The constrastive loss is derived by the cross-entropy loss (CE) between the normalised (norm) text encoder output and the normalised SCN encoder.

$$\mathcal{L}_{cl} = \text{CE}(\text{norm}(Z_t), \text{norm}(Z_s)) \tag{8}$$

At the same time, we include the training objective against the ground-truth label $y$, which is a linear interpolation of the text encoder and SCN encoder after transformation to the same dimension as the label space ($\tilde{\bullet}$) inspired by Lin et al. (2021).

$$Z = \frac{1}{2} \left( \text{softmax}(\tilde{Z}_s) + \text{softmax}(\tilde{Z}_t) \right) \tag{9}$$

$$\mathcal{L}_{label} = \text{CE}(Z, y) \tag{10}$$

The final loss function is the integration of the contrastive loss with the classification loss.

$$\mathcal{L} = \mathcal{L}_{label} + \eta \cdot \mathcal{L}_{cl} \tag{11}$$

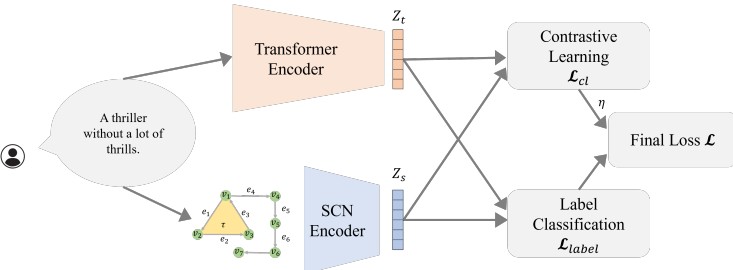

Figure 4: The Contrastive Learning with SCN (C-SCN) framework. The transformer encoder and SCN encoder will generate text representation $Z_t$ and document simplicial complex representation $Z_s$ respectively. The learned features will be used for contrastive learning by comparing themselves. Meanwhile, the two representations will contribute to the classification task with equal weights. As a result, the final loss $\mathcal{L}$ includes the contrastive loss $\mathcal{L}_{cl}$ and $\mathcal{L}_{label}$.

where $\eta$ is a control parameter.

## 4 EXPERIMENTS

### 4.1 DATASETS

The experiments are conducted on four datasets for short text classification tasks. The datasets are briefly introduced below, and a summary table is reported in Table 1. We adopt the same data preprocessing techniques as Wang et al. (2017) with slight modifications to include punctuation, keep the hashtag messages and add self-connection. **Twitter** (Bird et al. (2009)) is a binary classification dataset for sentiments "positive" and "negative" collected by Natural Language Toolkit. **MR** (Pang & Lee (2005)) contains movie review documents from Rotten Tomato with binary sentiment categories. **Snippets** (Phan et al. (2008)) contains Google web search text data with eight categories: "business", "computer", "health", "sports", "culture and art", "education and science", "engineering"and "politics and society". **StackOverflow** (Hamner et al. (2012)) contains question text from StackOverflow, and we choose the samples as Xu et al. (2017) for 20,000 questions with 20 categories.

Table 1: Summary statistics for text Datasets.

| Dataset | Twitter | MR | Snippets | StackOverflow |
|---|---|---|---|---|
| # Doc | 10,000 | 10,662 | 12,340 | 20,000 |
| # Train | 40 | 40 | 160 | 400 |
| Train ratio | 0.40% | 0.38% | 1.30% | 2% |
| # Tokens | 12,229 | 18,337 | 29,422 | 11,161 |
| Avg. Length | 9.3 | 20.4 | 18.0 | 9.3 |
| # Class | 2 | 2 | 8 | 20 |
| Avg. # 1-simplexes | 21.87 | 37.79 | 30.23 | 17.24 |
| Avg. # 2-simplexes | 0.24 | 0.74 | 3.24 | 0.21 |

### 4.2 BASELINE MODELS

We compare C-SCN with other various types of benchmark language models for short-text classification as reported by Liu et al. (2024).

*Traditional Language Models*: **TF-IDF** (Rajaraman & Ullman (2011)) refers to the term frequency-inverse document frequency, and it measures the importance of word tokens to the document. The features generated are passed in a support vector machine (SVM) (Crammer & Singer (2002)) for the classification task. **LDA** (Blei et al. (2003)) refers to Latent Dirichlet Allocation and extracts latent topics from the text through probabilistic models. The features are trained with SVM for short-text classification. **PTE** (Tang et al. (2015)) refers to Predictive Text Embedding, which utilises heterogeneous text networks for embeddings.

*Machine Learning Models*: **CNN** (Kim (2014)) refers to Convolutional Neural Networks with pre-trained GloVe word embeddings (Pennington et al. (2014)). **LSTM** (Liu et al. (2016)), which refers to Long-Short Term Memory, is trained GloVe embeddings. **BERT** (Devlin et al. (2019)), which refers to the Bidirectional encoder representations from transformers and its modified version **RoBERTa** (Zhuang et al. (2021)) leverages pre-training through self-supervised learning and could be fine-tuned to specific downstream tasks.

*Graph-based Language Models*: **TL-GNN** (Huang et al. (2019)) refers to text-level GNN, which adopts small windows for texts to focus on local features. **TextGCN** (Yao et al. (2019)), Text Graph Convolutional Network, constructs individual text graphs with document nodes based on word co-occurrences and word-document relations. **TextING** (Zhang et al. (2020)) adopts individual text graphs and inductively trains the model. **HyperGAT** (Ding et al. (2020)), Hypergraph Attention Networks enhances the expressive power of graphs on text classification by including high-order information and reducing computational resources needed for training. **STGCN** (Ye et al. (2020)), which refers to the short text graph convolutional network, integrates BERT and the bidirectional LSTM in graph models to enhance performance on short texts. **DADGNN** Liu et al. (2021), Deep Attention Diffusion Graph Neural Networks, applies attention diffusion and decoupling techniques targeting some limitations of GNN such as oversmoothing and restricted receptive field.

*Graph-based Models with external knowledge beyond documents or Contrastive Learning*: **STCKA** (Chen et al. (2019)) refers to Short Text Classification with Knowledge-powered Attention, which utilises attention mechanisms and entity conceptualisation to enhance text features. **HGAT** (Linmei et al. (2019)) is known as Heterogeneous Graph Attention Networks, and its enhanced version incorporates topic and entity beyond the texts for enriched graphs. **SHINE** (Wang et al. (2021)) is a hierarchical heterogeneous graph representation learning method for short text classification which executes entity and POS tagging for various types of node features. **NC-HGAT** (Sun et al. (2022)) integrates HGAT with neighbouring contrastive learning. **GIFT** (Liu et al. (2024)) is the graph contrastive learning for short text classification that employs SVD and $k$-means clustering methods in contrastive learning.

### 4.3 Implementation Details

Following with few-shot setting for short text classification framework (Sun et al. (2022); Wen & Fang (2023); Liu et al. (2024)), from each category, 20 samples are selected randomly to form the train set, another 20 samples are selected randomly to form the validation set, and the rest are included in the unseen test set. The 0-simplex embeddings are initialised with GloVe embeddings Pennington et al. (2014). The embedding matrices for 1-simplexes and 2-simplexes are randomly initialised and optimised to size 128. The learning rate is $1 \times 10^{-4}$, and the batch size is 128. A dropout rate of $50\%$ is implemented to reduce the complexity of the model and prevent overfitting problems. The model is trained with the PyTorch Geometric[1] package for 100 epochs with early stopping where the validation loss does not improve for ten epochs. The best weights are obtained from the model with the best validation accuracy. Cross-entropy loss is used with an Adam optimiser. The experiments are conducted ten times with NVIDIA RTX A6000 with 48GB of memory. We compare the results with strong baseline models with ten iterations of different training, validation and test sets. The average test accuracies and F1 scores are used for comparison.

## 5 Results and Discussion

### 5.1 Results

The experiment results are reported in Table 2. Compared with other competitive models, C-SCN has achieved the best test accuracies and F1 scores, indicating the model's ability to capture sentiments and sequential information in text documents.

We attribute the better performance to the following analysis. Firstly, we adopt SCN, a higher-order framework extending the expressive power of GNN. Features assigned to 0-simplexes, 1-simplexes and 2-simplexes could better represent the sentence structure and are generalised well across different contexts. The involvement of 1-simplexes and 2-simplexes in the message-passing mechanism

---

[1] https://pytorch-geometric.readthedocs.io/en/latest/index.html

Table 2: Results of test accuracy (%) and test F1-score (%) for short text classification where the best results based on 95% confidence in the pairwise $t$-tests are in bold, and the second-best results are underlined.

| Dataset | Twitter | | MR | | Snippets | | StackOverflow | |
|---|---|---|---|---|---|---|---|---|
| Metrics | F1 | Acc | F1 | Acc | F1 | Acc | F1 | Acc |
| TF-IDF | 53.62 | 52.46 | 54.29 | 48.13 | 64.70 | 59.17 | 59.19 | 59.06 |
| LDA | 54.34 | 53.97 | 54.40 | 48.39 | 62.54 | 56.4 | 60.19 | 59.52 |
| PTE | 54.24 | 53.17 | 55.02 | 52.62 | 63.10 | 59.11 | 62.56 | 61.32 |
| CNN | 57.29 | 56.02 | 59.06 | 59.01 | 77.09 | 69.28 | 63.75 | 61.21 |
| LSTM | 60.28 | 60.22 | 60.89 | 60.70 | 75.89 | 67.72 | 61.62 | 60.49 |
| BERT | 54.92 | 51.16 | 51.69 | 50.65 | 79.31 | 78.47 | 66.94 | 67.26 |
| RoBERTa | 56.02 | 52.29 | 52.55 | 51.30 | 79.55 | 79.02 | 69.91 | 70.35 |
| TL-GNN | 59.02 | 54.56 | 59.22 | 59.36 | 70.25 | 63.29 | 62.09 | 61.91 |
| TextGCN | 60.15 | 59.82 | 59.12 | 58.98 | 77.82 | 71.95 | 67.02 | 66.51 |
| TextING | 59.62 | 59.22 | 58.89 | 58.76 | 71.10 | 70.65 | 65.37 | 64.63 |
| HyperGAT | 59.15 | 55.19 | 58.65 | 58.62 | 70.89 | 63.42 | 63.25 | 62.10 |
| DADGNN | 59.51 | 55.32 | 58.92 | 58.86 | 71.65 | 70.66 | 66.26 | 65.10 |
| STCKA | 57.56 | 57.02 | 53.25 | 51.19 | 68.96 | 61.27 | 59.72 | 59.65 |
| STGCN | 64.33 | 64.29 | 58.25 | 58.22 | 70.01 | 69.93 | 69.23 | 69.10 |
| HGAT | 63.21 | 57.02 | 62.75 | 62.36 | 82.36 | 74.44 | 67.35 | 66.92 |
| SHINE | 72.54 | 72.19 | 64.58 | 63.89 | 82.39 | 81.62 | 73.05 | 72.73 |
| NC-HGAT | 63.76 | 62.94 | 62.46 | 62.14 | 82.42 | 74.62 | 67.59 | 67.02 |
| GIFT | 73.16 | 73.16 | 65.21 | 65.16 | 83.73 | 82.35 | 83.07 | 82.94 |
| SCN | 66.13 | 67.25 | 61.15 | 61.93 | 76.13 | 75.66 | 76.85 | 74.04 |
| C-SCN | **75.61** | **76.09** | **69.46** | **69.87** | **84.97** | **85.56** | **84.15** | **83.87** |

also expands the receptive fields of individual 0-simplexes where long-range information can be transmitted through shallow neural network layers, thereby enhancing the impact of 0-simplexes on the entire document. The self-attentive readout function connects 0-simplexes and 1-simplexes, creating expressive document-level summaries. This has promoted the SCN to perform the best in the benchmark datasets among the graph-based models without external information or contrastive learning. Secondly, the contrastive learning framework allows C-SCN to capture both structural and textual information in the few-shot setting. Both structural representation and sequential representation are treated as augmented views of each other. This has contributed to preventing helpful information from being removed, avoiding introducing noise or external information and combining the capabilities of both models.

In addition, we see that the large language models, such as BERT and RoBERTa, which leverage numerous pre-training, are not performing favourably with a few available labels. In contrast, graph-based models with external auxiliary knowledge or contrastive learning, including HGAT, SHINE, NC-HGAT, and GIFT, could achieve competitive results. External auxiliary knowledge, such as entity recognition and POS tagging, might help enrich the semantic and syntactic meaning of the original text. Still, it might be introducing extra noise and unnecessary messages to the text data, as shown in the deterioration of results from STGCN to HGAT. Furthermore, contrastive learning with perturbation of the graphs might inject misinformation about the text's meaning, explaining the difference between NC-HGAT and GIFT. Furthermore, introducing the global network within the small train set where the connectivity or clustering effect is explored might not be significant. This could explain why our model could outperform SHINE and GIFT.

## 5.2 ABLATION STUDIES

Ablation studies are conducted to remove individual components to verify the capability of higher-order simplexes and contrastive learning in enhancing text understanding. The results are reported in Table 3. It is observed that removing contrastive learning deteriorates the results for both SCN and BERT. Regarding higher-order simplexes, the removal of any component might deprecate the test accuracies and F1 scores across all datasets. Moreover, the inclusion of 1-simplexes followed

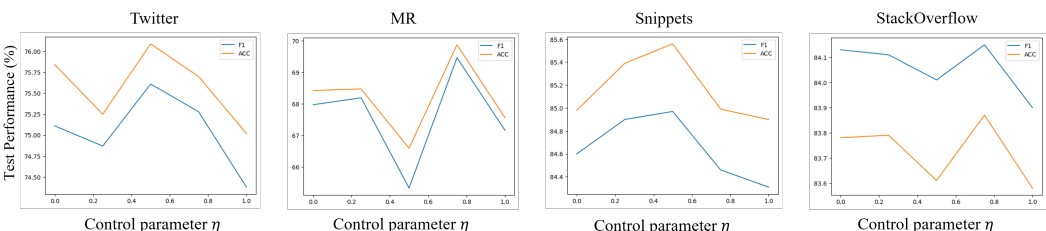

Figure 5: Hyperparameter $\eta$ sensitivity across different datasets.

by the inclusion of 2-simplexes improves the results respectively, highlighting the importance of higher-order simplicial complexes in document understanding.

Table 3: Results of test accuracy for ablation studies. "C-SCN - 0-simplex" means the 1-simplexes and 2-simplexes are both removed in the model, whereas "C-SCN - 1-simplex" refers to the removal of 2-simplexes from the model.

| Dataset | Twitter | | MR | | Snippets | | StackOverflow | |
|---|---|---|---|---|---|---|---|---|
| Metrics | F1 | Acc | F1 | Acc | F1 | Acc | F1 | Acc |
| BERT | 54.92 | 51.16 | 51.69 | 50.65 | 79.31 | 78.47 | 66.94 | 67.26 |
| SCN | 66.13 | 67.25 | 61.15 | 61.93 | 76.13 | 75.66 | 76.85 | 74.04 |
| C-SCN - 0-simplex | 74.50 | 74.78 | 67.48 | 68.27 | 84.58 | 85.13 | 82.79 | 82.36 |
| C-SCN - 1-simplex | 74.91 | 75.41 | 68.54 | 68.77 | 84.75 | 85.32 | 83.08 | 82.58 |
| C-SCN | **75.61** | **76.09** | **69.46** | **69.87** | **84.97** | **85.56** | **84.15** | **83.87** |

The hyperparameter sensitivity of $\eta$ is investigated across different datasets, and the results are visualised in Figure 5. The control parameter $\eta$ indicates the weights of contrastive loss in the model training process. We could observe that there are various types of impact on test performance. In general, the test performance varies between the value 0 (no contrastive loss) and 1 (higher weight of contrastive loss), while $\eta = 1$ results in lower performance compared to the case of no contrastive loss. One explanation for such variation could be the need to balance the focus between achieving the correct label and synchronising model weights between SCN and the transformer model. In our experiments, a grid search is conducted for the best performance for the best $\eta$ values.

## 6 CONCLUSION

In conclusion, we propose Contrastive Learning with Simplicial Convolutional Networks (C-SCN), which incorporates higher-order information for sequence analysis and is applied in short text classification tasks. The model constructs document simplicial complexes and develops a convolutional network to incorporate the higher-order simplexes' message passing with a self-attention readout. Furthermore, we integrate the transformer model to generate augmented views in the contrastive learning framework. Extensive experiments that simulate the lack of label situation in a few-shot setting indicate that our model leverages advantages from both structural and sequential representation, learns long-range information and enhances textual understanding with contextualised 1-simplexes and 2-simplexes during training.

In the future, we would like to explore the interpretability of higher-order simplexes and their roles in text understanding. The impact of the number of 1-simplexes and 2-simplexes on the performance of C-SCN is also worth attention, and it could be more inspected within the context of longer documents. Leveraging SCN's expressiveness in sequential analysis could have more applications in other fields, such as recommender systems and process mining.

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

## A  EFFICIENCY STUDIES

In order to study computational efficiency with the inclusion of higher-order objects, we compute the number of trainable parameters, as shown in Table 4.

Table 4: Number of trainable parameters.

|  | Twitter | MR | Snippets | StackOverflow |
|---|---|---|---|---|
| SCN – 0-simplex | 3,654,154 | 6,120,154 | 9,446,428 | 3,969,676 |
| SCN – 1-simplex | 3,671,050 | 6,137,050 | 9,463,324 | 3,986,572 |
| SCN | 3,672,202 | 6,138,202 | 9,464,476 | 3,987,724 |
| C-SCN – 0-simplex | 113,236,620 | 115,702,620 | 119,029,668 | 113,554,464 |
| C-SCN – 1-simplex | 113,253,516 | 115,719,516 | 119,046,564 | 113,571,360 |
| C-SCN | 113,255,568 | 115,720,668 | 119,047,716 | 113,572,512 |

The time to complete training and evaluation after ten iterations in seconds is also included for analysis, as shown in Table 5.

It is observed that when adding 1-simplexes and 2-simplexes to SCN step-by-step, the average number of trainable parameters increases by 0.18%, and the time increases by 10.55% on average. For C-SCN, the number of trainable parameters increases by less than 0.1% on average, and the time for training increases by 3.32% on average. The results demonstrate the computational efficiency of our model involving higher-order complexes in representation learning.

Table 5: Time to complete training and evaluation after ten iterations.

|  | Twitter | MR | Snippets | StackOverflow |
|---|---|---|---|---|
| SCN – 0-simplex | 639 | 730 | 1,009 | 1,448 |
| SCN – 1-simplex | 679 | 743 | 1,235 | 1,728 |
| SCN | 750 | 798 | 1,279 | 1,956 |
| C-SCN – 0-simplex | 1,015 | 1,033 | 1,943 | 3,080 |
| C-SCN – 1-simplex | 1,025 | 1,152 | 1,970 | 3,191 |
| C-SCN | 1,040 | 1,197 | 2,007 | 3,384 |

# B ADDITIONAL RESULTS FOR ABLATION STUDIES

## B.1 COMPARED WITH CONTEXTUAL EMBEDDINGS IN CONTRASTIVE LEARNING

Instead of fixed GloVe embedding for word nodes, we compare the results with contextual embeddings (Cont. Emb.) from the BERT model in the following table.

Table 6: Results of test accuracy to compare with the separate contrastive loss.

| Dataset | Twitter | | MR | | Snippets | | StackOverflow | |
|---|---|---|---|---|---|---|---|---|
| Metrics | F1 | Acc | F1 | Acc | F1 | Acc | F1 | Acc |
| BERT | 54.92 | 51.16 | 51.69 | 50.65 | 79.31 | 78.47 | 66.94 | 67.26 |
| C-SCN - Cont. Emb. | 74.60 | 75.01 | 50.46 | 55.34 | 83.31 | 83.96 | 82.53 | 83.00 |
| C-SCN | **75.61** | **76.09** | **69.46** | **69.87** | **84.97** | **85.56** | **84.15** | **83.87** |

It is observed that C-SCN with fixed embeddings achieves better results than the one with contextual embeddings. One explanation could be the limited number of higher-order objects formed with contextual embeddings. 0-simplexes (nodes), which refer to the same word, will not be seen as the same 0-simplex at different locations in the document with contextual embeddings. This will lead to no 2-simplexes formed in the document since one 0-simplex will not be connected again within the text, limiting the expressiveness of structural representations of the higher-order objects.

## B.2 SEPARATE CONTRASTIVE LOSS FROM THE OBJECTIVE

To evaluate the effectiveness of optimising the contrastive loss and objective function together, experiments to separate the two losses (Sep. Loss) are also conducted. The contrastive loss is first minimised for 100 epochs without labels, and the loss against the final label is optimised with early stopping. The result is shown in the following table.

Table 7: Results of test accuracy to compare with and without GRU.

| Dataset | Twitter | | MR | | Snippets | | StackOverflow | |
|---|---|---|---|---|---|---|---|---|
| Metrics | F1 | Acc | F1 | Acc | F1 | Acc | F1 | Acc |
| BERT | 54.92 | 51.16 | 51.69 | 50.65 | 79.31 | 78.47 | 66.94 | 67.26 |
| C-SCN - Sep. Loss | 67.54 | 68.35 | 53.61 | 56.79 | 64.69 | 64.8 | 27.01 | 29.78 |
| C-SCN | **75.61** | **76.09** | **69.46** | **69.87** | **84.97** | **85.56** | **84.15** | **83.87** |

It is observed that with limited training samples (20 samples from each category), pre-training with a contrastive loss followed by supervised training does not help the model improve. In detail, the separate contrastive loss could improve BERT's performance in binary classification in Twitter and MR datasets. In contrast, it worsens the performance in multi-label classification, and the most deterioration is from the StackOverflow dataset, which has 20 categories.

### B.3 THE ROLE OF GRU IN THE MESSAGE FUNCTION

We adopted GRU as the UPDATE function to control the amount of information from the previous step and aggregated neighbourhood information. This is achieved through the reset gate and the reset gate structure in GRU. In contrast, we study the role of GRU by comparing the performance if we remove GRU as the UPDATE function and replace it with the sum aggregation (SUM).

Table 8: Results of test accuracy to compare with contextual embeddings.

| Dataset | Twitter | | MR | | Snippets | | StackOverflow | |
|---|---|---|---|---|---|---|---|---|
| Metrics | F1 | Acc | F1 | Acc | F1 | Acc | F1 | Acc |
| SCN - SUM | 62.3 | 63.21 | 54.99 | 56.23 | 77.06 | 77.05 | 76.02 | 73.73 |
| SCN | 66.13 | 67.25 | 61.15 | 61.93 | 76.13 | 75.66 | 76.85 | 74.04 |
| C-SCN - SUM | 74.01 | 74.45 | 55.51 | 58.26 | 77.38 | 77.92 | 78.73 | 77.46 |
| C-SCN | **75.61** | **76.09** | **69.46** | **69.87** | **84.97** | **85.56** | **84.15** | **83.87** |

One challenge we observed without GRU was the overfitting issue on the train set across different datasets. The results deteriorated when we removed GRU from SCN and C-SCN respectively, illustrating the importance of GRU in the message-passing mechanism for higher-order complexes.

### B.4 PSEUDO-CODE FOR C-SCN

We include the pseudo-code for C-SCN to enhance the reproducibility.

---

**Algorithm 1:** Algorithm Pseudo Code for C-SCN.

---

**Input:** Text data with *words, punctuations and label* as shown in Figure 3.

**Simplicial Complex Construction**
Tokenised words from the document data $\mathbf{x}_{\text{doc}}$;
Tokenised unique 0-simplex $\sigma_0 \in \mathcal{K}_0$;
1-simplex indices following the chronological order of tokens $\sigma_1 \in \mathcal{K}_1$;
1-simplex features tokenised to one of the types: *forward, backward, self-loop*;
2-simplex features tokenised by the flow directions of components;
```
/* Add higher-order simplexes if needed.                    */
```

**Model Construction**
**Parameters:**
Embedding matrices for 0-simplexes, 1-simplexes and 2-simplexes: $\mathcal{E}_0$, $\mathcal{E}_1$ and $\mathcal{E}_2$;
Number of layers: $L$;
Message-passing mechanism for 0-simplexes and 1-simplexes following Equation 2 and
  Equation 3: $\text{MP}_0$, $\text{MP}_1$;
Attention mechanism for 0-simplexes and 1-simplexes following Equation 4: $\text{Attn}_0$, $\text{Attn}_1$;
Activation function: $\phi$;
Transformer model: $\text{Trans}_t$;
Linear layers that process the output of the transformer and the SCN to the label space: $\text{MLP}_t$,
  $\text{MLP}_s$.
**Initialise features:**
  $\mathbf{h}_{\sigma_0}^{(0)} = \mathcal{E}_0(\sigma_0) \, \forall \, \sigma_0 \in \mathcal{K}_0$; $\mathbf{h}_{\sigma_1}^{(0)} = \mathcal{E}_1(\sigma_1) \, \forall \, \sigma_1 \in \mathcal{K}_1$; $\mathbf{h}_{\sigma_2}^{(0)} = \mathcal{E}_2(\sigma_2) \, \forall \, \sigma_2 \in \mathcal{K}_1$.
**for** $\ell = 1$ *to* $L - 1$ **do**
  $\mathbf{h}_{\sigma_1}^{(\ell)} = \text{MP}_1(\mathbf{h}_{\sigma_1}^{(\ell-1)}, \mathbf{h}_{\sigma_2}^{(\ell-1)})$;
  $\mathbf{h}_{\sigma_0}^{(\ell)} = \text{MP}_0(\mathbf{h}_{\sigma_0}^{(\ell-1)}, \mathbf{h}_{\sigma_1}^{(\ell-1)})$.
$\mathbf{h}_{\mathcal{K}}^{(L)} = \sum_{\sigma_1 \in \mathcal{K}_1} \text{Attn}_1(\mathbf{h}_{\sigma_1}^{(L)}) \oplus \sum_{\sigma_0 \in \mathcal{K}_0} \text{Attn}_0(\mathbf{h}_{\sigma_0}^{(L)})$;
$Z_t = \text{MLP}_t(\text{Trans}_t(\mathbf{x}_{\text{doc}}))$; $Z_s = \text{MLP}_s(\mathbf{h}_{\mathcal{K}}^{(L)})$;
$Z = \frac{1}{2}(\text{softmax}(Z_s) + \text{softmax}(Z_t))$;
**return** $\hat{y} = Z, Z_s, Z_t$.

---

