# OpenReview forum: "Contrastive Learning with Simplicial Convolutional Networks for Short-Text Classification"
_ICLR.cc/2025/Conference — Submitted to ICLR 2025_

### Official Review · Reviewer_z4Dx · 2024-11-01

**Soundness:** 3
**Presentation:** 2
**Contribution:** 3
**Rating:** 6
**Confidence:** 3

**Summary:**

This paper presents a novel way to construct contrastive learning negative and positive pairs through topological deep learning lens (using higher order of transformations, in this case, 0-, 1- and 2-simplex) with applications to text classification task in NLP domain. The framework SCN performs reasonably well with respect to other contrastive learning baselines, this is further improved by additionally using contextual representations from BERT with additional self-contrast. Resulting model C-SCN archives promising results in various benchmarks.

**Strengths:**

This paper has the following novelty:

1. for contrastive learning: the perspective of topological deep learning is introduced
2. for text classification task: a solid way to construct higher order transformations in the context of text is introduced
3. the paper provide enough evidence through experiments to show their method works

**Weaknesses:**

1. I think the explanations of section 3.1 could be introduced with a bit clearer motivations and justifications. Especially figure 1, this should be explainable more carefully on how this DAG or computational graph is constructed with some solid examples (for example, I will start with a real data example and show step by step how the graph is composed). (I also could not find code so can not verify how this is done). Compare to how this graph is constructed, understanding of what 0-, 1- and 2-simplex means is more straightforward.

**Questions:**

1. Why not use the contextual embedding in the first place in the SCN model? It is clear to me from your experiments, the biggest gain in terms of performance happens after the self-contrastive with the BERT embedding. I think some of the other benchmark methods use contextual embedding to initialise their contrastive learning framework directly. Maybe the authors could discuss the tradeoffs or potential advantages of their approach compared to using contextual embeddings from the start.

2. Is there anything more recent work in the literature related to your work? Given the fact so many people are focusing on graph contrastive learning, it is a bit surprising there is not much follow up. Also, topological deep learning is also a very active research field but there is no more recent references? I will advise authors to include a brief discussion of the most recent developments in graph contrastive learning and topological deep learning, and how their work relates to or differs from these recent advances.

3. Could please add a section on discussing how constructing contrastive learning graph impact the task performance and how decisions should be made? It seems to me there are many inductive bias/human judgement to be involved and this even makes comparing different methods tricky.

---

### Official Review · Reviewer_E2fa · 2024-11-04

**Soundness:** 3
**Presentation:** 3
**Contribution:** 3
**Rating:** 6
**Confidence:** 4

**Summary:**

This paper proposes a novel text classification method based on graph deep learning. The authors introduce the use of Simplicial Convolutional Networks (SCN) to model the structural relationships within text in an inductive way. In this model, 0-simplexes represent word nodes, 1-simplexes represent edges between words, and 2-simplexes represent three word nodes forming a triangle. Node features are derived from pre-trained word embeddings. The embeddings of nodes and edges are obtained through a message-passing mechanism applied to the three types of simplexes. Finally, a READOUT function is used to obtain the representation of the entire document for classification. The author also proposed a contrastive learning framework that combines Transformer and SCN representations as an extension. The authors validate the performance of SCN on four short text datasets, demonstrating that it performs better than other graph-based deep learning methods. Additionally, the authors conduct ablation studies on the three types of simplexes.

**Strengths:**

The paper is well-written and easy to follow.

The idea of constructing simplicial complexes is interesting.

The proposed SCN performs better or comparably to other graph-based deep learning methods.

**Weaknesses:**

The motivation for introducing contrastive learning seems somewhat weak, as it may not be applicable to scenarios without label scarcity, potentially reducing the proposed method's impact.

The distinction between the contrastive learning method proposed by the author and that of Wen & Fang (2023) is not very clear. It is uncertain whether better results can be achieved solely by comparing different contrastive learning methods under identical conditions.

**Questions:**

Could the author provide some examples where other methods misclassified but the proposed method classified correctly, and explain on what kind of short texts it has an advantage?

---

### Official Review · Reviewer_jJFN · 2024-11-05

**Soundness:** 3
**Presentation:** 3
**Contribution:** 2
**Rating:** 5
**Confidence:** 4

**Summary:**

In this paper, the authors observe that existing short-text classification methods generally exhibit certain limitations, such as the potential introduction of unnecessary semantic noise during contrastive learning data augmentation, and the constraints of graph models in capturing higher-order features. Accordingly, this paper proposes Contrastive Learning with Simplicial Convolutional Networks (C-SCN), a short-text classification framework that integrates higher-order features with contrastive learning. Specifically, in the SCN module, documents are modeled as simplicial complexes, from which useful information is extracted through a message-passing mechanism. The features of 0-simplexes and 1- simplexes are then aggregated via a global self-attention mechanism to form the final graph representation of the SCN module. Simultaneously, BERT is employed as a text encoder in the transformer module to generate textual representations. C-SCN processes the graph representation from the SCN module and the textual representation from the transformer module in parallel, enabling joint optimization for contrastive learning and classification tasks. Based on this, a final loss function is proposed, incorporating both classification and contrastive losses. Experimental results on four datasets and eighteen baselines demonstrate the effectiveness of the proposed method.

**Strengths:**

(1)	This paper observes that some previous short-text classification methods employing graph models face limitations in capturing higher-order features, such as constrained group-wise interactions due to the number of layers. In response, this paper proposes an SCN message-passing mechanism model that constructs and leverages simplicial complexes to obtain higher-order graph representations for text classification tasks.
(2)	To enhance performance, this paper extends SCN to contrastive learning by processing the graph representation from the SCN module and the textual representation from the transformer module in parallel.
(3)	This paper conducts extensive experiments on four datasets with eighteen baselines. The baseline methods encompass a broad range of models, including traditional language models, machine learning models, graph-based language models, and graph-based models utilizing external knowledge beyond documents or contrastive learning, ensuring comprehensive baseline selection. Additionally, the experimental section includes essential ablation studies and parameter sensitivity analyses.

**Weaknesses:**

Overall, the innovation of the paper appears somewhat limited. The definition of simplicial complexes and the details of contrastive learning loss can be found in Bodnar et al (2021) and Wen & Fang (2023), respectively. This paper seems to merely combine these two components.
	A repeatedly emphasized premise in this paper is that "the text data could be enhanced with external auxiliary information that might introduce noise to the sparse text data." However, the paper does not explicitly address how this noise is managed.
	Is there a typographical error in σ_k^' on line 201? My understanding is that it should be corrected to σ_k.
	Bodnar et al (2021) appears to define a more general simplicial complex that incorporates higher-order information. Moreover, the description of adjacent simplexes in lines 97-104 has already been formally defined in Bodnar et al (2021). Why does this paper discard the more general, higher-dimensional simplicial complex from Bodnar et al (2021) in favor of a highly similar redefinition? What is the rationale behind this choice? Are there any fundamental differences between these two complexes?
	The novelty in the contrastive learning component appears insufficient, as the definitions of contrastive loss and classification loss seem to have counterparts in Wen & Fang (2023). Has this paper introduced any improvements that are better suited to the short-text classification scenario based on these existing definitions?
	This paper selects a large number of baseline methods for comparison; however, only one baseline, GIFT, is from 2024, while the others are primarily from around 2020 or earlier. Please include more recent baseline methods for experimentation.
	The results of the ablation study show that the main performance improvement comes from contrastive learning, while the addition of 2-simplices does not lead to a significant performance boost. Is the increased time complexity introduced by 2-simplices justified?

**Questions:**

Please refer to the Weaknesses.

---

### Official Review · Reviewer_ARBN · 2024-11-08

**Soundness:** 3
**Presentation:** 2
**Contribution:** 2
**Rating:** 5
**Confidence:** 4

**Summary:**

This paper is in line with text classification. It addresses the ineffectiveness of existing contrastive learning in modeling auxiliary information. Specifically, it proposes a novel document simplicial complex construction based on text data for a higher-order message-passing mechanism. It develops a simplicial complex representation for text sentences based on the directed word co-occurrence. These designs help improve short text classification performance.

**Strengths:**

- This paper is well written and easy to follow.
- The proposed simplicial complex feature is interesting. Building graphs from such 0-simplexes, 1-simplexes, and 2-simplexes can effectively capture intrinsic higher-order structural information within words.
- A well-designed message-passing mechanism is proposed to gather information from neighboring simplexes and co-boundary adjacent simplexes.

**Weaknesses:**

- The experiments and discussions presented are insufficient, and it would be beneficial to include a more comprehensive analysis, such as an efficiency study.
- The current baselines are outdated; incorporating more recent and robust baselines, such as large language models (LLMs), would help demonstrate the effectiveness of the proposed model.
- In Figure 5, the observed changes in performance are significant, yet there is limited discussion about these changes. Currently, the contrastive and classification losses are optimized jointly. It would be nice to explore the effects of training these losses separately to better understand their impact on performance.
- How does the efficiency of modeling the proposed simplex features compare to other graph baselines? It would be better to provide an efficiency study.

**Questions:**

### Questions

- I am curious about why BERT and RoBERTa underperform LSTM and CNN on the Twitter and MR datasets. This seems counterintuitive. I reviewed the BERT results on the MR dataset from paper [1], where BERT and RoBERTa significantly outperform CNN and LSTM. Could you verify your experimental settings to ensure the accuracy of the reported results?
- In Equation 4, could you explain why do you use GRU to update information? Additionally, it would be helpful to include a citation for GRU.
- The 2-simplex may be sparser than the 0- and 1-simplex. From Table 3, we can see that omitting the 2-simplex has little impact on performance. However, I am concerned that introducing the 2-simplex may increase complexity and negatively affect efficiency.


### Suggestions:
- L69: “’what” -> ``what''
- L114: integrate -> integrates
- L302: MLP• -> MLP(•)


### Reference

- [1] Karl, F., & Scherp, A. Transformers are short text classifiers: a study of inductive short text classifiers on benchmarks and real-world datasets. CoRR abs/2211.16878 (2022). URL: https://doi. org/10.48550/arXiv, 2211.

---

### Meta-Review · Area_Chair_Wq2E · 2024-12-22

**Metareview:**

### Claims and Findings:
 - This paper introduces a short-text classification framework that constructs a simplicial complex representation for text sentences to capture higher-order structural information among words. Additionally, it integrates contrastive learning to enhance performance.
### Strengths:
   - The proposal of simplicial complexes to capture higher-order features in text data is reasonable and convincing.
  - The overall design of the framework is sound.
### Weaknesses:
   - The paper does not clearly explain the novelty of the proposed method, particularly in relation to previous relevant work such as Bodnar et al. (2021) and Wen & Fang (2023), although the authors addressed this in their replies to the reviewers.
   - The baselines used for comparison are somewhat outdated, and it is unclear how the proposed method performs against modern LLMs across a wide range of scenarios.
   - The writing could be improved in terms of language, organization, clarity, and comprehensiveness.

### Reasons for Decision:
   - Based on the identified weaknesses.

**Additional Comments On Reviewer Discussion:**

While the rebuttal clarified the paper's novelty relative to previous work, it remains uncertain how the method compares to contemporary LLMs. Overall, the paper demonstrates potential but would benefit from another round of revisions to include additional experiments and comparisons, as well as to enhance its clarity and quality.

---

### Decision · Program_Chairs · 2025-01-22

Reject